# Biofilm Spreading by the Adhesin-Dependent Gliding Motility of *Flavobacterium johnsoniae*. 1. Internal Structure of the Biofilm

**DOI:** 10.3390/ijms22041894

**Published:** 2021-02-14

**Authors:** Keiko Sato, Masami Naya, Yuri Hatano, Yoshio Kondo, Mari Sato, Keiji Nagano, Shicheng Chen, Mariko Naito, Chikara Sato

**Affiliations:** 1Department of Microbiology and Oral Infection, Graduate School of Biomedical Sciences, Nagasaki University, Nagasaki 852-8588, Japan; mnaito@nagasaki-u.ac.jp; 2Health and Medical Research Institute, National Institute of Advanced Industrial Science and Technology (AIST), Tsukuba 305-8566, Japan; masami.naya@gmail.com (M.N.); iyuhri@gmail.com (Y.H.); ma-satou@aist.go.jp (M.S.); 3Department of Pediatric Dentistry, Graduate School of Biomedical Sciences, Nagasaki University, Nagasaki 852-8588, Japan; yosioji@nagasaki-u.ac.jp; 4Department of Microbiology, Health Sciences University of Hokkaido, 1757 Kanazawa, Tobetsu-cho, Ishikari-gun, Hokkaido 061-0293, Japan; knagano@hoku-iryo-u.ac.jp; 5Department of Clinical and Diagnostic Sciences, School of Health Sciences, Oakland University, 433 Meadow Brook Road, Rochester, MI 48309, USA; schen5@oakland.edu

**Keywords:** extracellular polymeric matrix, extracellular fibers, vesicle, bacterial clusters, cell-to-cell connections, transmission electron microscopy

## Abstract

The Gram-negative bacterium *Flavobacterium johnsoniae* employs gliding motility to move rapidly over solid surfaces. Gliding involves the movement of the adhesin SprB along the cell surface. *F. johnsoniae* spreads on nutrient-poor 1% agar-PY2, forming a thin film-like colony. We used electron microscopy and time-lapse fluorescence microscopy to investigate the structure of colonies formed by wild-type (WT) *F. johnsoniae* and by the *sprB* mutant (Δ*sprB*). In both cases, the bacteria were buried in the extracellular polymeric matrix (EPM) covering the top of the colony. In the spreading WT colonies, the EPM included a thick fiber framework and vesicles, revealing the formation of a biofilm, which is probably required for the spreading movement. Specific paths that were followed by bacterial clusters were observed at the leading edge of colonies, and abundant vesicle secretion and subsequent matrix formation were suggested. EPM-free channels were formed in upward biofilm protrusions, probably for cell migration. In the nonspreading Δ*sprB* colonies, cells were tightly packed in layers and the intercellular space was occupied by less matrix, indicating immature biofilm. This result suggests that SprB is not necessary for biofilm formation. We conclude that *F. johnsoniae* cells use gliding motility to spread and maturate biofilms.

## 1. Introduction

*Flavobacterium johnsoniae* is a Gram-negative rod-shaped aerobic bacterium commonly found in soil and fresh water. The cells of this bacterium move rapidly over solid surfaces by gliding motility and thus form thin spreading colonies on agar [1,2]. This characteristic motility is shared by many other members of the *Bacteroidetes* phylum that lack bacterial motility organelles, such as flagella or pili. Instead, their movements depend on a complex motility-specific apparatus that has been intensively studied in *F. johnsoniae*, which is a model system [3,4].

SprB with a mass of 669 kDa is a cell surface adhesin component of the motility machinery of *F. johnsoniae* and forms filaments on the cell surface [5,6,7]. SprB filaments were observed on WT cells using ammonium molybdate staining and transmission electron microscopy (TEM), but were not present on *sprB* mutant (Δ*sprB*) cells [6]. SprB appears to be propelled rapidly along a closed helical loop track by the motility machinery in a process that requires a proton motive force, resulting in gliding of the cell [6,7]. Cryo-electron tomography revealed filaments and outer membrane-associated patches near the base of the outer membrane of WT cells, but these were not observed when the nonmotile *gldF* mutant cells were similarly examined [8]. SprB is secreted and translocated to the cell surface by the Bacteroidetes-specific type IX secretion system (T9SS), which is intertwined with the gliding motility machinery [9,10,11,12,13,14,15]. Therefore, T9SS mutants influenced gliding and caused nonspreading colonies on both agar surfaces.

Colony spreading of *F. johnsoniae* is influenced by modifications of the motility machinery complex, such as SprB deficiency, and by environmental factors, such as temperature and nutrient availability [16]. When cultured on nutrient-poor agar medium, gliding cells of *F. johnsoniae* form thin film-like spreading colonies [2]. Although these exhibit apparent gross similarities to typical bacterial biofilms, it remains unclear whether they are organized and constructed in the same way.

A microbial biofilm is a sessile community of surface-attached microorganisms embedded in self-produced extracellular polymeric matrix (EPM) that grows on various solid surfaces with a fluid interface [17,18]. Biofilms are involved in many chronic infectious diseases such as periodontal disease, bacterial endocarditis and bacterial osteomyelitis [19,20,21]. In the *Flavobacterium* genus, the widely distributed fish pathogens *Flavobacterium columnare* and *Flavobacterium psychrophilum* form biofilms when they colonize the gill [22,23,24]. Cai et al. showed that EPM and water channels are present in mature biofilms of *F. columnare* under aqueous condition [25]. As revealed by scanning confocal laser microscopy, in biofilms formed by *F. psychrophilum*, most live bacterial cells are found in the deeper and intermediate layers, while dead cells predominate in the upper layer [22].

Some members of the *Bacteroidetes* phylum exhibiting gliding motility cause infectious diseases in humans and fish [26]. *Capnocytophaga canimorsus*, a commensal bacterium in dog and cat mouths, can cause rare severe human infections [27,28]. Zhang et al. showed that colony morphology is associated with the virulence of *F. columnare*. *F. columnare* forms three colony morphotypes (rhizoid, rough and soft), but only the rhizoid morphotype is virulent in rainbow trout [29,30]. Furthermore, T9SS deletion mutants of *F. columnare* exhibit reduced virulence in zebrafish [30]. Mutants of *gld* genes necessary for the gliding of *F. psychrophilum* also exhibit reduced virulence in zebrafish [31,32]. Shrivastava et al. showed that cells of the gliding bacterium *Capnocytophaga gingivalis* present in the human oral microbiome carry polymicrobial cargoes, including nonmotile species, to new locations, where they form colonies [33]. These data suggest that the gliding motility of bacteria influences their virulence.

It is known that gliding of *F. johnsoniae* cells on agar is needed for colony spreading, but structural analyses of differences between cells and interactions of cells in a spreading wild type colony and in a nonspreading *sprB* mutant colony have not been analyzed in detail.

Here, we investigated the detailed structure of the spreading colony in wild-type *F. johnsoniae* (WT) and an *sprB* deletion mutant CJ1922 (Δ*sprB*) on nutrient-poor agar media using time-lapse fluorescence microscopy and TEM [34]. The WT cells were found to be embedded in an EPM that contained a thick filamentous network and vesicles, indicating biofilm formation. The bacterial cells at the extending tip of WT colonies were densely surrounded by budding vesicles and many secreted vesicles. In nonspreading Δ*sprB* colonies, the cells were tightly packed in layers and the intercellular space was occupied by less matrix, indicating immature biofilm. We showed that the motility, population kinetics, matrix production and cell localization within the biofilm were influenced by the cell surface adhesin SprB-dependent gliding machinery.

## 2. Results

### 2.1. WT F. johnsoniae Colonies Spread on Nutrient-Poor Agar Surfaces

To investigate *F. johnsoniae* colony spreading, we compared the behavior of the wild-type (WT) strain and an *sprB* deletion mutant strain (Δ*sprB*) [34]. A 1 μL drop of washed *F. johnsoniae* UW101 WT cells was initially inoculated on nutrient-poor 1% agar PY2 medium (1% A-PY2), and the cells were incubated at room temperature (RT) (23–24 °C) for 5 days. During this period, the WT cells grew and spread radially from all edges of the inoculation spot at the same speed to form a large circular colony (Figure 1a, left). In contrast, the Δ*sprB* mutant CJ1922 colony did not spread (Figure 1a, right) [5,34]. The radius of the developing WT colony depended on the time after incubation (Appendix A).

### 2.2. Movement of Single Bacterial Cells in the WT Colony Spread on 1% A-PY2

To visualize the movement of single bacterial cells at the colony edge, a 1:100 mixture of WT cells with and without cytoplasmic GFP expression was inoculated in the center of an agar plate, incubated for 24 h, and visualized by time-lapse fluorescence microscopy (Appendix A) [16,35]. This video (at 300× the speed of standard time-lapse fluorescence microscopy imaging) is shown in Appendix A. On 1% A-PY2, the colony spread out from the inoculated spot. At the leading edge of the spreading colony, a small cell cluster at the tip of a branch moved outwards and was followed by other cells (Figure 1b, upper panel, Appendix A). This suggested cell-to-cell connections. Furthermore, cells often moved along the path on the agar surface used by the preceding cell clusters (Appendix A), suggesting the existence of a track formed by the leading cells controlling the movement of the following cell clusters.

### 2.3. Static Single Bacterial Cells in the ΔsprB Mutant Colony on 1% A-PY2

The same experiment was performed for *sprB* deletion mutant cells. The Δ*sprB* colony did not spread, and curved lines of cells producing cytoplasmic GFP were apparent within the colony (Figure 1a,b and Appendix A). The video of Δ*sprB* colony (at 300× the speed of standard time-lapse fluorescence microscopy imaging) is shown in Appendix A. Because *F. johnsoniae* cells divide along a single axis, it is reasonable to propose that the cells in the line all originated from the same cell. The movement of individual cells in the colony was not detectable.

### 2.4. WT F. johnsoniae Colonies Form a Biofilm on Nutrient-Poor Agar Surfaces

To investigate the structure of a spreading colony more precisely, the WT colony was aldehyde-fixed, embedded in Epon, and vertically sectioned parallel to the direction of spreading (Appendix A). The approximately 70 nm thick, 1 mm wide, and 0.5 mm long sections were stained with heavy metals and inspected by TEM. At the bottom of the WT colony spreading on 1% A-PY2, i.e., close to the agar layer, cells were loosely packed and sparsely embedded in a low electron density matrix (Figure 2). Bacterial cells were positioned in different directions. Above the bottom layer, there were fewer cells.

### 2.5. Bacteria at the Advancing Front of WT Colonies Secreted Many Vesicles

To observe the edge structure of WT colonies, the concentric WT colony formed on 1% A-PY2 was divided into six equal regions at the colony margin (<0.5 mm from the edge) (Figure 3a,b). Epon embedding and TEM revealed that the layered coat on the surface of cells found in all regions showed irregular undulations (Figure 3c). The intercellular space was occupied by a matrix containing thin extracellular fibers (3–7 nm in diameter) interspersed with secreted vesicles (Figure 3c). The bacteria advancing the furthest in the colonies were more densely surrounded by budding vesicles and many secreted vesicles (30–50 nm in diameter) (Figure 3c, image 1–2 and 1′). In the inner regions of the colony, each bacterial cell was surrounded by only a few smaller vesicles (Figure 3c, image 3–6 and 3′). The densities of vesicles attached to the cells in the regions 1–2 were higher than those in the regions 3–6 (Figure 3d). Some round vacancies that could be gas bubbles were visible at the front edge of the colony (Figure 3b1,3b2 and 3c1). All these results suggested biofilm formation.

### 2.6. WT Cells form a Queue at the Advancing Vertical Top of the Colony

Upon detailed observation of the advancing top of the WT cell layer, we found strings of cells protruding upwards away from the main colony, presumably due to the pile-up induced by the increase in cell population (Figure 4). As shown in Figure 4a, the strings could be classified into four distinct regions, I to IV, that extend perpendicular to the culture substrate. No bacterial cells were found in the region farthest from the agar surface (region I), but fiber-like structures were rather uniformly distributed throughout the area (Figure 4bI). We suggest that these fibers were secreted from the leading edge of the cell population. Region II contained bacteria that were presumably close to the top of the colony. A thin channel without any thin extracellular fibers was observed in the bottom half of this region (Figure 4a). The bacteria in region II were densely surrounded by budding vesicles and many isolated vesicles (Figure 4bII,c). In contrast, each bacterial cell was surrounded by only a few relatively smaller vesicles in region III and by even fewer vesicles in region IV, the least distal upper region of the colony, closest to the bottom layer (Figure 4a,b). Thin extracellular fibers were viewed from top to bottom from region I through region IV and were fully distributed from the front of the cell population. The filamentous background in region II, which lies under region I, was traversed from top to bottom by a filament-less channel that contained many cells (Figure 4a). More vesicles were attached to the cells in region II than those in region III and IV (Figure 4c).

### 2.7. Biofilm Maturation Depends on the Cell-Surface Adhesin SprB

Next, we investigated whether the motility adhesin SprB is required for biofilm formation in *F. johnsoniae*. To this end, we evaluated the spreading phenotype of an *sprB* deletion mutant on 1% A-PY2. Deletion of *sprB* prevented *F. johnsoniae* colonies from spreading on 1% A-PY2; the cells only grew within the small inoculation circle (Figure 1a, right). This growth pattern was in stark contrast to the behavior of the WT strain, which spread radially from all edges of the inoculation spot on the agar surface (Figure 1a, left). To investigate the role of SprB in detail, as for the WT, a Δ*sprB* colony was embedded in Epon, sliced, and observed by TEM (Figure 5). Unlike the WT cells, the mutant cells were crowded and formed three distinct layers on the 1% A-PY2 medium (Figure 5). Higher magnification images revealed that the Δ*sprB* cells had a smooth rod-like shape (Figure 6a–h).

In the base layer, cells were tightly packed on the agar surface (Figure 6b–d,g–h) and grouped in preferred orientations due to the high cell density; variations in the length of the similarly shaped cells within a group were relatively small (Figure 6b–d,g–h). In the middle of the base layer, the intercellular space was occupied by many large and small vesicles, extracellular filaments and a few secreted bacterial cytoskeletal-like structures, but the separation between cells was clearly small (Figure 6c,g). In some places, lysed cells that maintained gross cell shape of membranes were observed (decreased contrast in cytoplasm) (Figure 6c,g). These suggests that the biofilm formed was immature.

A large cluster of cells was observed above the base layer, the region we refer to as the 2nd layer (Figure 5 and Figure 6a,b,e,f). In contrast to the tightly packed base layer, curved and straight rod-shaped cells of various lengths were adjacent to one another, resulting in less dense communities and loss of directionality (Figure 5, 2nd layer, Figure 6e,f). Some cells spread out from the cluster, forming a less populated 3rd layer further towards the top (Figure 5, 3rd layer, Figure 6a,e). Similar to the WT cells, the space between the Δ*sprB* cells in the 2nd and 3rd layers was occupied by a substance containing thin extracellular fibers and vesicles (Figure 6a,b,e,f). Within the 2nd layer, each Δ*sprB* cell possessed a thick surface coat structure (Figure 7, lower), similar to WT (Figure 7, upper).

### 2.8. Surface Connecting Structure of F. johnsoniae Cells in Biofilm

Within the biofilm, each WT cell was oriented in various directions and contained a thick layered surface coat structure (Figure 7). The layered coat structure had regular undulations along the long and short axes of the cells and contacted the coat of neighboring cells (Figure 7). The coat structure was not observed when the bacterial cells isolated from the colony grown on 1% A-PY2 were washed with washing buffer (10 mM Tris-HCl pH 7.5) prior to aldehyde fixation (Figure 8). The remaining outer membrane surrounding the WT cells was undulated (Figure 8, left), as observed in Figure 7. Such undulation of outer membrane was also observed for the cells from Δ*sprB* colony (Figure 8a, right).

## 3. Discussion

The attachment of colonies to a solid surface is a complex process mediated by cell-cell interactions. For *F. johnsoniae*, this process was affected by the cell surface components and environmental factors, such as the nutrient supply and moisture (Figure 1, Figure 2, Figure 3, Figure 4 and Figure 5) [16]. Biofilms of many bacteria, such as *P. aeruginosa* and *E. coli*, are surface-attached microbial communities composed of cells embedded in an extracellular polymeric matrix (EPM) [36,37]. These matrices are composed of polysaccharides, extracellular DNA and protein structures such as curli, fimbriae and pili [38,39,40,41]. The biofilm formed by *P. aeruginosa*, which exhibits flagellum-mediated swimming motility and surface-associated swarming and twitching motilities, is established in five main stages: (i) attachment, involving adhesion of bacteria to the substrate; (ii) cell–cell adhesion; (iii) early development of biofilm architecture; (iv) maturation of biofilm architecture; and (v) dispersion of single dissociated cells from the biofilm [42,43,44].

As reported earlier [2], WT *F. johnsoniae* cells grew and formed a characteristic large circular colony on 1% A-PY2 (Figure 1a). It is not clear how the cells were distributed and connected three-dimensionally in the spreading colony. Here, we studied the precise structure of the SprB-dependent spreading and the Δ*sprB* nonspreading colonies using electron microscopy (EM) (Figure 2, Figure 3, Figure 4, Figure 5 and Figure 6). This showed that cells were embedded in a self-produced EPM, forming a biofilm, while they divided and glided on the agar medium. Consequently, the biofilm of the colony spread from the central region outwards.

TEM of colony sections of WT *F. johnsoniae* revealed that the space between the cells was occupied by an EPM containing fibers and vesicles (Figure 3 and Figure 4). These fibers, 3–7 nm in diameter, were somewhat thicker than those observed in biofilms formed by *Staphylococcus aureus* [45] and *P. acnes* [46]. This might reflect the tough structure required for the mobile nature of the *F. johnsoniae* biofilm, in contrast to the static biofilms of *S. aureus* and *P. acnes.* In the WT *F. johnsoniae* colony spreading on 1% A-PY2, the internal space was sparsely populated by cells. At its marginal top, extracellular fibers were distributed from the leading edge of the cell population (Figure 3b). In the matrix, a cell trajectory was clearly identified by the presence of a filament-less channel, through which following cells could move (Figure 4a, upper left). These observations suggest that secretion of EPM preceded the migration of *F. johnsoniae* cells. In total, colony expansion seems to reflect the morphological, behavioral and physiological characteristics of the cells in the biofilm colony (Figure 4).

Attachment to a solid surface is an important first step in biofilm formation [47,48,49]. Because gliding motility is a movement that allows bacteria to stay in contact with the solid surface, it appears that *F. johnsoniae* cells, partly with the assistance of adhesins such as SprB, can glide on the agar surface covered with secreted EPM while secreting vesicles, thus extending the biofilm. The bacteria in the advancing fronts toward horizontal (Figure 3a, region 1–2) or upward direction (Figure 4a, region II) were more densely surrounded by budding vesicles and many secreted vesicles in the colony (Figure 3d, image 1–2 and Figure 4c, region II). The abundant vesicle secretions might support EPM formation because vesicles could include or attach to proteins, including chaperons, contributing to EPM formation.

WT cells grew and formed a large circular colony on 1% A-PY2, whereas the Δ*sprB* mutant formed nonspreading colonies. Although the Δ*sprB* cells were densely packed, the intercellular space was occupied by a substance containing thin extracellular fibers and vesicles (Figure 5 and Figure 6). Consequently, the cell surface adhesin SprB has a role in the expansion of *F. johnsoniae* biofilm, although SprB is not required for biofilm formation. This is in good accordance with a previous study showing that the *P. aeruginosa* biofilm architecture formed by a flagella- and type IV-pili-double mutant was different from that formed by WT [49]; flagella- and twitching-motility have roles in shaping the biofilm architecture, although they are not necessary for biofilm formation.

The volume of extracellular matrix of the Δ*sprB* mutant colony was significantly less than that of the WT colony (Figure 6). Our data suggest that gliding movement is associated with the volume of extracellular matrix and thus affects the maturation of biofilm architecture. Recently, we investigated the SprB-independent colony spreading of *F. johnsoniae* on soft agar containing glucose [16]. The extracellular matrix of the colony contained a network of extracellular fibers and many secreted vesicles, which is similar with that on 1% A-PY2 [16,50].

Death of cells was observed in the tightly packed region of the *F. johnsoniae* Δ*sprB* mutant colony that grew on 1% A-PY2 (Figure 5 and Figure 6, the base layers). Such death might reflect social apoptosis, including activities forming cell-free channels, to allow external nutrition to diffuse into the biofilm. Consequently, the thick fiber-containing EPM might include double-stranded DNA filaments and extracellular polysaccharides, as observed in the biofilms of *S. aureus* and *P. acnes* immersed in aqueous liquid by atmospheric scanning electron microscopy (ASEM) [45,46]. Further experiments are required to understand the water-rich architecture of *F. johnsoniae* biofilms, for example, using ASEM that enabled observations of various water-rich phenomena of organic and inorganic substances at high resolution [16,51,52]. The undulations observed for the layered surface coats of *F. johnsoniae* cells in biofilm has a possibility to be artefacts, due to the dehydration pretreatment of the Epon-embedded TEM. It should be also addressed using the liquid-phase ASEM.

The results reported here shed light into the liquid-phase biofilm structures formed at the interfaces between air and water, and also between water and solid substrate. The various electron microscopy and optical microscopy techniques used in this studies will accelerate study of such biofilms.

## 4. Materials and Methods

### 4.1. Bacterial Strain and Biofilm Cultivation

*F. johnsoniae* strains were grown in casitone-yeast extract (CYE) medium at 24 °C (Becton, Dickinson and Co., Franklin Lakes, NJ, USA). The details of the bacterial strains and plasmids used are shown in Table 1. For selection and maintenance of antibiotic-resistant *F. johnsoniae* strains, antibiotics were added to the medium at the following concentrations: streptomycin, 100 μg/mL; erythromycin, 100 μg/mL.

To observe colony spreading, *F. johnsoniae* WT and *sprB* deletion mutant CJ1922 (Δ*sprB*) cells were grown in CYE medium at 27 °C with shaking (175 rpm) overnight. The cells were collected as a pellet by centrifugation at 800× *g* for 10 min, at 22 °C. The pellet was resuspended in the same volume of washing buffer (10 mM Tris-HCl pH 7.4) by vortexing, and the suspension was centrifuged at 800× *g* for 10 min at 22 °C. These steps were repeated twice. The cells were spotted onto peptone yeast (PY2) agar medium (peptone and yeast extract, Becton, Dickinson and Co. and agar, Ina Food Industry Co., Ltd., Nagano, Japan) in a dish 9 cm in diameter and incubated at 24 °C [15]. Construction of an *F. johnsoniae* strain expressing GFP was carried out as follows: After the mating of *E. coli* S17-1 λ*pir* carrying pFj29 with *F. johnsoniae* WT (CJ1827) and CJ1922, an Em^r^ transconjugant was obtained [34,53].

### 4.2. Fixation

For Epon embedding and TEM, spreading colonies on agar medium and cultured bacterial cells were fixed with 1% paraformaldehyde and 3.5% glutaraldehyde in 0.1 M phosphate buffer (PB) (pH 7.4) at room temperature (RT) for 3 h and further with 1% osmic acid in the same buffer at 4 °C for 1 h.

### 4.3. Epon Embedding and Sectioning

Fixed colonies were dehydrated through a gradient series of alcohol at RT and embedded in Epon 812. Ultrathin sections (70 or 400 nm thick) were cut parallel to the colony spreading direction and perpendicular to the agar medium surface. This allowed both spreading across the surface of the agar medium and any penetration into the agar to be monitored. A Leica Ultracut UCT microtome (Leica, Wetzlar, Germany) was employed. A series of ultrathin sections was cut at RT and collected on EM grids.

### 4.4. TEM Imaging

Epon sections mounted on grids were stained with uranyl acetate (UA) and lead citrate (LC) and observed with an H7600 TEM (Hitachi, Tokyo, Japan) at 80 kV.

### 4.5. Quantification of Vesicles

The number of vesicles attached to the outer circumference of bacteria was manually counted and divided by the outer circumference to get vesicle density (vesicles/μm). The circumference of the cells was measured using ImageJ (National institute of Health). Thirty-three cells were analyzed for each region.

## Figures and Tables

**Figure 1 ijms-22-01894-f001:**
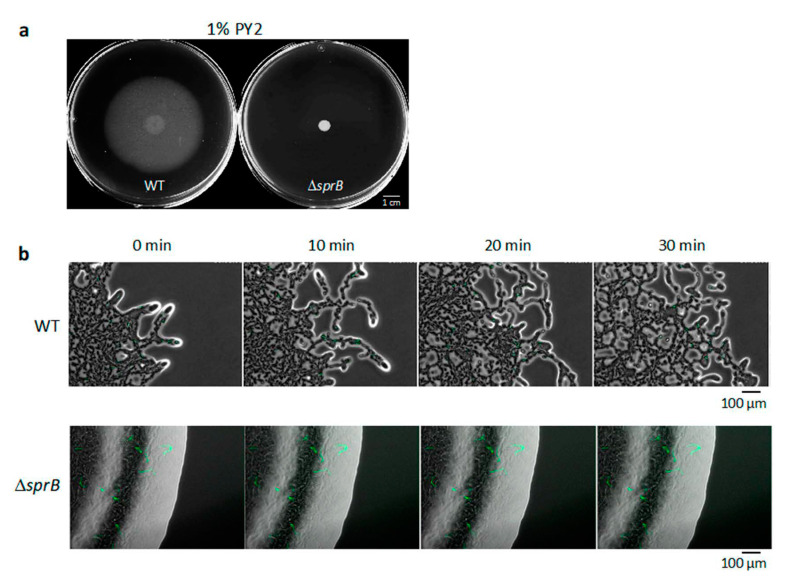
Colony spreading of WT and Δ*sprB* mutant cells on 1% A-PY2. (**a**) Images of the colonies spreading on 9-cm diameter dishes (5 days). (**b**) Behavior of bacteria in spreading of WT and Δ*sprB* colonies. Cells expressing GFP in their cytoplasm were added to the inoculated bacterial solution at a concentration of 1%, and movement of the bacterial cells at the colony edge was monitored. Images were recorded by fluorescence microscopy at 30-s intervals for 30 min. Upper panel: WT. At the leading edge of a spreading WT colony on 1% A-PY2, a small cell cluster moved outward at the tip of a branch. Lower panel: Δ*sprB* mutant. At the leading edge of a nonspreading Δ*sprB* colony, curved lines of static cells expressing GFP were observed and were interpreted to be cells connected in line. Time is shown at the top left. Selected frames showing the edge of each bacterial colony during 10 min of video captures (see Appendix A).

**Figure 2 ijms-22-01894-f002:**
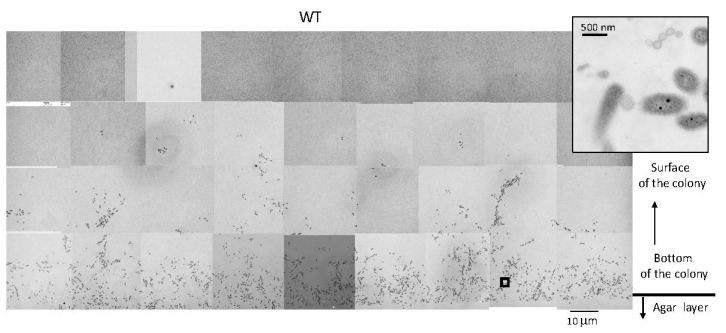
TEM of spreading colonies of WT *F. johnsoniae* on 1% A-PY2. WT colonies were aldehyde-fixed and embedded in Epon resin. Then, 70 nm-thick sections were cut parallel to the direction of colony growth and perpendicular to the surface of the agar medium (Appendix A), stained with metal solutions and examined by TEM. Overview of the sections examined. High-magnification images of cells in Figure 2 are shown in Figure 7. Inset is the enlargement of the square in the left panel.

**Figure 3 ijms-22-01894-f003:**
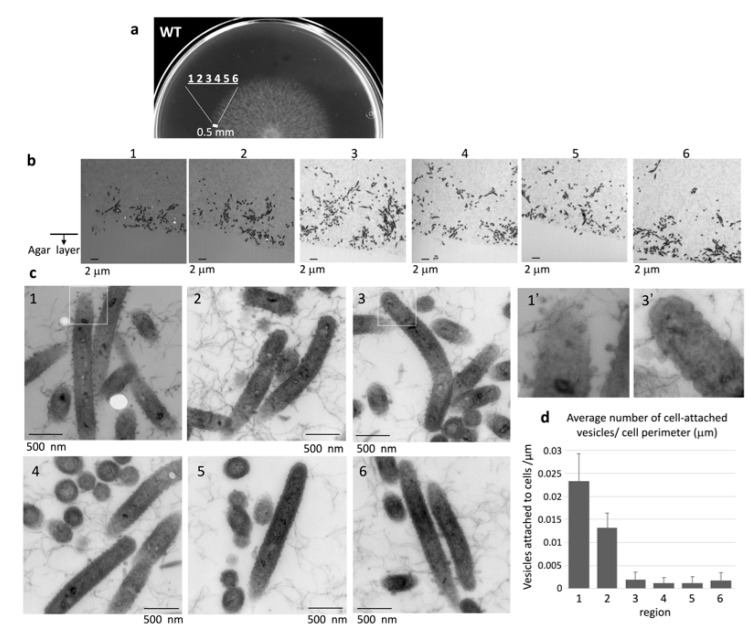
TEM of the extending edge of a WT colony on 1% A-PY2. (**a**) Overview of the sampled region of the spreading colony. (**b**) Images from regions 1–6 are indicated in Figure 3a. The bacterial layer on the agar surface is shown, with the cross-section along the axis of spreading and perpendicular to the agar surface (Appendix A). The vacant area at the bottom of each image is the agar layer. (**c**) (1–6) Higher magnification images of the corresponding panels in Figure 3b. (1′) and (3′) are 3× enlarged images of the squares in (1) and (3), respectively. (**d**) The density of cell-attached vesicles per perimeter of cells (vesicles/mm) was measured. The areas around 33 cells were analyzed for each region. More vesicles were attached to the cells in region 1 and 2 than those in region 3, 4, 5 and 6 (Figure 3b,c). Symbols on the graph represent the average density, and the error bars correspond to the SD.

**Figure 4 ijms-22-01894-f004:**
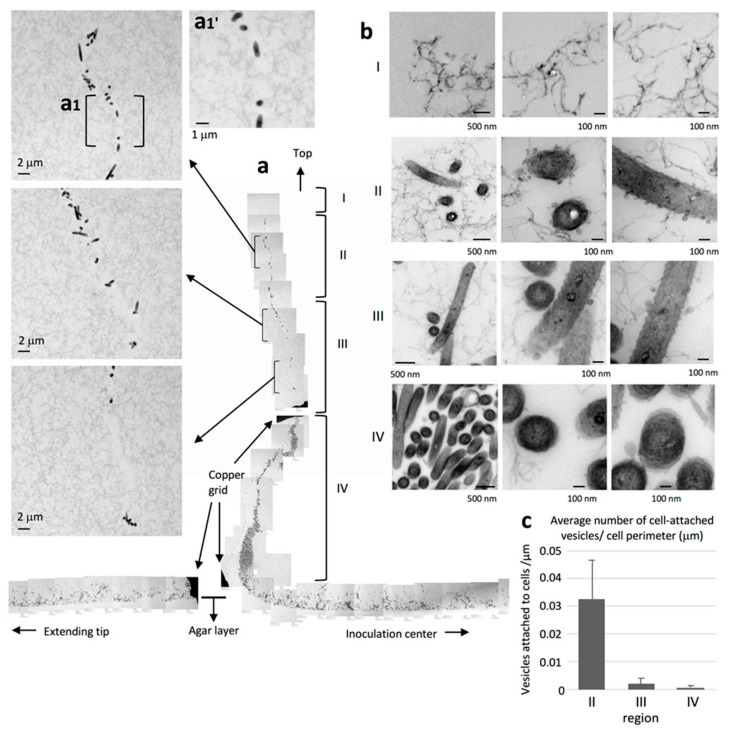
Strings of clustered WT bacteria protrude upwards, away from the main colony and the agar surface. WT colonies grown on 1% A-PY2 were aldehyde-fixed and embedded in Epon resin, and 70 nm sections were cut perpendicular to the surface of the agar medium and examined by TEM. Four distinct regions could be discerned. (**a**) Overview of the sections. Left panels: Higher-magnification images of region II and III in Figure 4a. Top panel: (a1′) Enlarged image of the area (a1) in Figure 4a. The filamentous background is traversed from top to bottom by a filament-less channel that contains many cells. (**b**) High-magnification images of regions I–IV. Region I is the highest in the marginal zone of the main colony and furthest from the agar plate. No bacteria are apparent; only fibers are observed at the leading edge of the biofilm. Region II: bacteria with many large vesicles and fibers surrounding the cells are observed at the top of the colony. Region III: bacteria with a few large vesicles surrounding them are observed. Region IV, just above the bottom bacterial layer: bacteria with a few small vesicles surrounding them are observed. Overall, the large intercellular space rich in filaments and vesicles in regions I–IV suggests mature biofilm formation in the WT colony. (**c**) The density of cell-attached vesicles per perimeter of cells (vesicles/mm) was measured. More vesicles were attached to the cells in region II than those in region III and IV (Figure 4b). Symbols on the graph represent the average density, and the error bars correspond to the SD.

**Figure 5 ijms-22-01894-f005:**
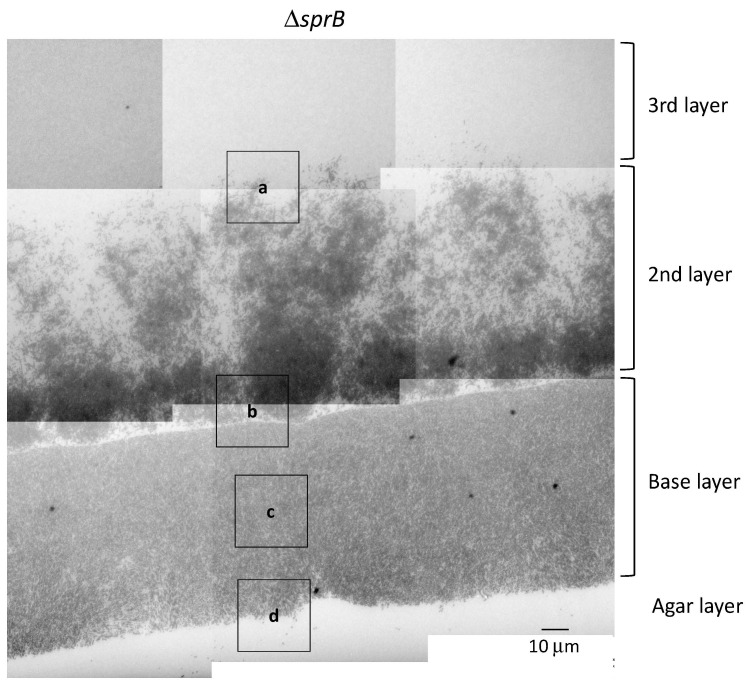
TEM of spreading colonies of the *F. johnsoniae* Δ*sprB* mutant on 1% A-PY2. Colonies of Δ*sprB* were aldehyde-fixed and embedded in Epon resin. Then, 70 nm-thick sections were cut parallel to the direction of colony growth and perpendicular to the surface of the agar medium (Appendix A), stained with metal solutions and examined by TEM. Overview of the sections examined. High-magnification images of cells pictured in Figure 5 are shown in Figure 6 and Figure 7 (second row).

**Figure 6 ijms-22-01894-f006:**
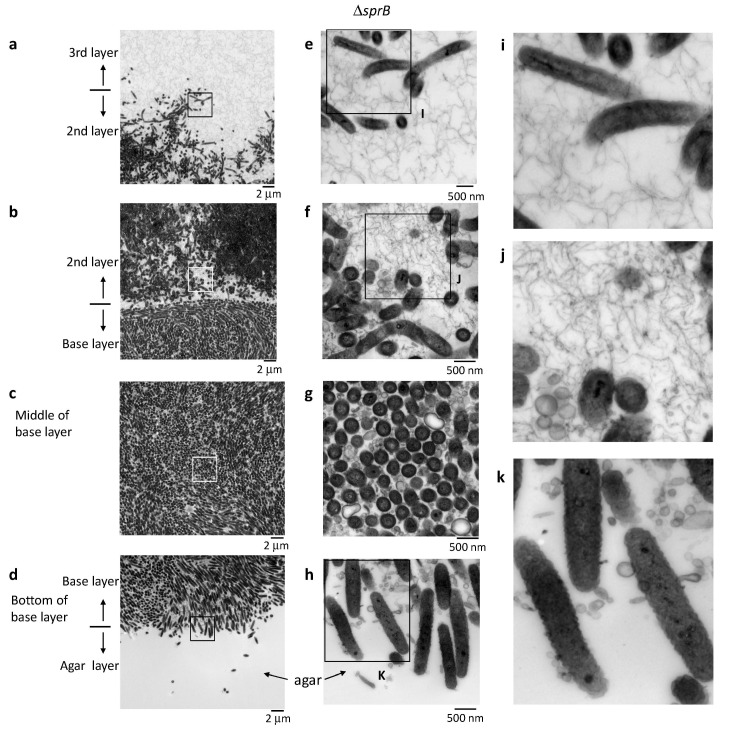
Colony spreading of the Δ*sprB* on 1% A-PY2. Higher-magnification images of cells shown in Figure 5. The corresponding position of each panel is indicated by the squares in Figure 5. (**a**–**d**) Left panels: Images of the spreading colony showing the bottom of the 3rd layer/top of the 2nd layer, the bottom of the 2nd layer/top of the base layer, middle of the base layer, and the bottom of the base layer/top of the agar, as indicated. (**e**–**h**) Middle panels: Higher magnification images of the regions shown on the left. (**i**–**k**) Right panels: Enlargement of the annotated regions shown on the left. The smaller intercellular space containing fewer filaments and vesicles suggests that the biofilm formed by the Δ*sprB* is immature.

**Figure 7 ijms-22-01894-f007:**
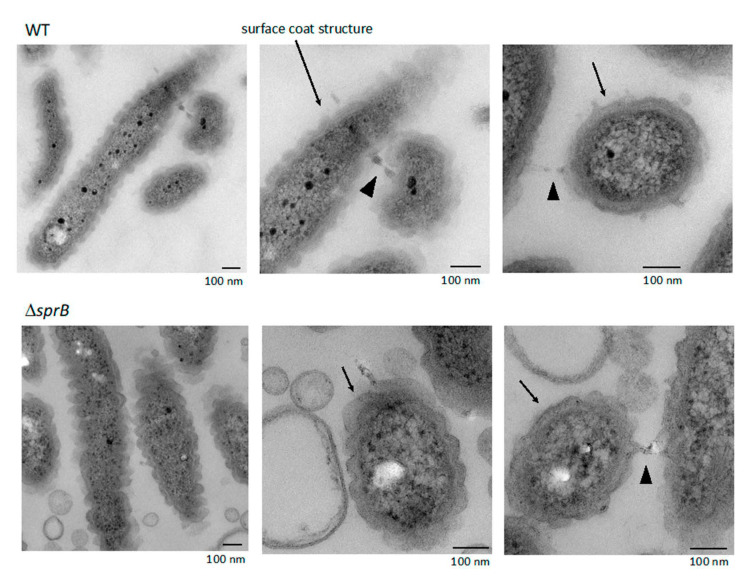
WT and Δ*sprB* mutant cells on 1% A-PY2 embedded in Epon imaged by TEM. Embedded cells were 70 nm thin-sectioned. High-magnification images of cells pictured in Figure 1a. Each cell in the bacterial layer on the agar surface had a thick layered coat structure (arrows). The coat structure had regular undulations along the long and short axes, and was often in contact with the coat of the neighboring cells (arrowheads).

**Figure 8 ijms-22-01894-f008:**
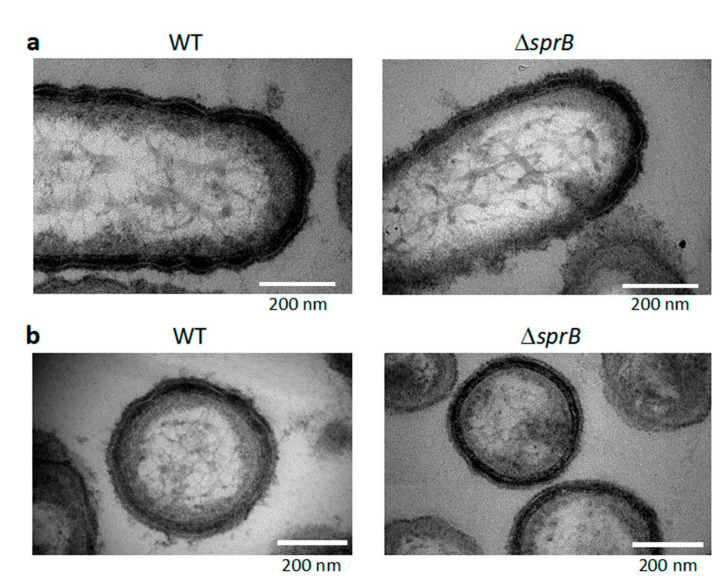
TEM images of buffer-washed WT and Δ*sprB* cells isolated from their colonies formed on the agar. Cells were isolated from a few mm-zone from the edges of WT and Δ*sprB* colonies formed on 1% A-PY2, washed with the washing buffer, and aldehyde fixed. After dehydration, the cells were embedded in Epon. The samples were 70 nm thick thin-sectioned, stained, and imaged by TEM. (**a**) Thin-sections along the long axis of cells. (**b**) Thin-sections along the short axis of cells. After the washing, surface coat structure was not observed. The undulations of the remaining outer membrane were still observed for both WT and Δ*sprB* cells.

**Table 1 ijms-22-01894-t001:** Bacterial strains and plasmids used in this study.

Strain	Description	Ref.
*E. coli strain*		
S17-λpir	RP4-2-Tc::Mu *aph*::Tn7 *recA*, Sm^r^	[53]
*F. johnsoniae strain*		
CJ1827	*rpsL2*, Sm^r^	[34]
CJ1922	*sprB*, Sm^r^	[34]
*F. johnsoniae plasmid*		
pFj29	Ap^r^ Em^r^, Expression vector carrying with *ompA* promoter and *gfp*	[35]

## Data Availability

Not applicable.

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
