# Peer review of "Biofilm Spreading by the Adhesin-Dependent Gliding Motility of Flavobacterium johnsoniae. 1. Internal Structure of the Biofilm"

_ijms, 2021, doi:10.3390/ijms22041894_

Round 1

Reviewer 1 Report

This study provides a detailed look at the ultrastructure of F. johnsoniae biofilms both along the surface of the agar and perpendicular to it. Interesting insights include the density of cells at different points, the presence of vesicles and extracellular polymeric matrix (EPM) and the formation of EPM-free migration channels. The SprB mutant being deficient in motility forma a morphologically different biofilm.

Main points

  1. Fig 3. The main text analysis of figure 3 can be improved. The difference between Fig 3c image 1 and image 2 concerning the density of vesicles is not convincing. It may be better to group image 1 and 2 in line 145 and compare to image 3-6 in line 146. This would be consistent with the quantitation shown in Fig 3d. Fig 3d should be described and the number of cells quantitated should be given in the legend text. The enlarge portions 1’ and 3’ are helpful and should be referred to in the text to strengthen your interpretation.
  2. Lines 127-131. This description claims the appearance of a matrix of extracellular fibers which are not visible. “Intracellular” should be “intercellular”? Perhaps the description of the matrix should be deferred to Fig 3. What is the vertical “plate”?
  3. Lines 230-33 and 252-255. The interpretation that the mutant OM is smoother is not readily observable from the images provided.
  4. A very interesting feature of the biofilms are the migration tracks devoid of EPM. Are similar structures observed in biofilms produced by other motile bacteria? Please discuss. Also, the idea in line 282-3 that EPM secretion was preceded by migration needs to be clarified. It might be just the language, but the results suggest that EPM secretion happens first and when the bacteria glide through it, the EPM gets pushed aside? More detail on this and perhaps a figure describing the model would be helpful.
  5. Previous studies of FJ ultrastructure should be included in the introduction. The description of individual cells, such as their undulating surface must be compared to previous literature including cryo-EM studies which preserve the native ultrastructure (e.g Liu et al, J Bacteriol. 2007 Oct;189(20):7503-6.). Are the undulations real or is it possible that they are an artefact of the EM procedure that include dehydration?
  6. The conclusions in lines 27-29 and 79-83 should be made more consistent with the results and discussion. The differences observed between WT and mutant are not consistently reported.

Minor points

  1. The EM methodology needs to be reviewed. There are several terms like nanogold, ASEM and immunolabeling under “4.2 fixation” which don’t appear to be used in the actual study. Therefore, the actual method used is very unclear.
  2. The gldK mutant is not used in this study – please remove from acknowledgement
  3. The method for quantifying vesicles (for Fig 3d, 4c) should be described in the methods
  4. Fig 4a. It is unclear where the three main images come from. There are lines for the top and bottom image suggesting that the top image comes from the top-most image that has bacteria in section II, while the bottom-most image appears to come from the lowest image in section III. Is that correct? But where does the middle image come from? It is also confusing that there are two labels “Top surface” and “furthest from agar surface” – are these describing the same location? Also, the axis along the surface of the agar can be better labeled.
  5. The abbreviation for micro (mu) has been replaced by alpha in many of the figures.
  6. Fig 5. The inset labels should just be “a”, “b”, “c” and “d” as they refer to Fig 6 not Fig 5.
  7. Line 305-6. Dead cells are mentioned here in the discussion but were not mentioned in the original description of the figures. How is it known that they are dead?
  8. Line 258. The meaning of this statement is unclear “For F. johnsoniae, this process was affected by the cell surface conditions”
  9. The last phrase in lines 315-6 needs revising.
  10. Reference #24 is currently missing details of journal, submission status etc
  11. Reference #20 is currently missing details of submission status

Author Response

Dear Reviewer1,

International Journal of Molecular Sciences,

January 27, 2021

Manuscript ID: ijms-1058259

Thank you for your email on 7-January-2021 regarding our manuscript, “Biofilm spreading by the adhesin-dependent gliding motility of Flavobacterium johnsoniae. 1. Internal structure of biofilm”.

We are very grateful for yourowncomments and the reviewer’s comments, and would like to express our cordial thanks for the valuable suggestions for our paper. We have carefully revised the text, and send you the revised version (attached), and detailed our point-by-point responses to each issue raised. Changes in the text are indicated by yellow highlights.

We hope that the revised manuscript is now suitable for publication in the International Journal of Molecular Sciences.

Sincerely,

Keiko Sato, DDS, PhD

Department of Microbiology and Oral Infection,

Nagasaki University Graduate School of Biomedical Sciences

1-7-1 Sakamoto, Nagasaki, Nagasaki 852-8588, Japan

Tel: +81-95-819-7649. Fax: +81-95-819-7650.

Chikara Sato, Ph.D.

Health and Medical Research Institute, National Institute of Advanced Industrial Science and Technology (AIST)

Umezono 1-1-4, Tsukuba, Ibaraki 305-8568 Japan

Tel: +81-29-861-5562; Fax: +81-29-861-6478

Responses to reviewer1’s comments

First of all, we would like to express our cordial thanks to referee 1 for his/her insightful comments, which encouraged us and helped us to significantly improve the paper.

This study provides a detailed look at the ultrastructure of F. johnsoniae biofilms both along the surface of the agar and perpendicular to it. Interesting insights include the density of cells at different points, the presence of vesicles and extracellular polymeric matrix (EPM) and the formation of EPM-free migration channels. The SprB mutant being deficient in motility forma a morphologically different biofilm.

Response: Thank you for your high evaluation for our paper. The comments are very encouraging for us.

Main points

Comment#1: Fig 3. The main text analysis of figure 3 can be improved. The difference between Fig 3c image 1 and image 2 concerning the density of vesicles is not convincing. It may be better to group image 1 and 2 in line 145 and compare to image 3-6 in line 146. This would be consistent with the quantitation shown in Fig 3d. Fig 3d should be described and the number of cells quantitated should be given in the legend text. The enlarge portions 1’ and 3’ are helpful and should be referred to in the text to strengthen your interpretation.

Response: The reviewer's comment is correct.We have changed from “The bacteria advancing the furthest in the colonies were more densely surrounded by budding vesicles and many secreted vesicles (30–50 nm in diameter) (Fig. 3c, image 1). In the inner regions of the colony, each bacterial cell was surrounded by only a few smaller vesicles (Figs. 3c, image 2-6 and 3d).” to “The bacteria advancing the furthest in the colonies were more densely surrounded by budding vesicles and many secreted vesicles (30–50 nm in diameter) (Fig. 3c, image 1-2 and 1’). In the inner regions of the colony, each bacterial cell was surrounded by only a few smaller vesicles (Figs. 3c, image 3-6 and 3’). The densities of vesicles attached to the cells in the regions 1-2 were higher than those in the regions 3-6 (Fig. 3d).”in page 4 line 145-150 in the revised manuscript.

We have changed from “(d) The density of cell-attached vesicles per perimeter of cells (vesicles/mm) was measured.” to “(d) The density of cell-attached vesicles per perimeter of cells (vesicles/mm) was measured. The areas around 33 cells were analyzed for each region.” in page 5 line 158-159 in the legend text of the revised manuscript.

We have corrected from “(1’) and (3’) are enlarged images of the squares in (1) and (2), respectively” to “(1’) and (3’) are enlarged images of the squares in (1) and (3), respectively.” in page 5 line 157-158 in the revised manuscript.

According to the changes in the results, we have added the following sentences “The bacteria in the advancing fronts toward horizontal (Fig. 3a, region 1-2) or upward direction (Fig. 4a, region II) were more densely surrounded by budding vesicles and many secreted vesicles in the colony (Fig. 3d, image 1-2 and Fig. 4c, region II).”in page 10-11 line 291-294 in the discussion of the revised manuscript.

We have changed from “Abundant vesicle secretion, especially at the colony edge (Fig. 4), might support EPM formation because vesicles could include or attach to proteins, including chaperons, contributing to EPM formation.” to “The abundant vesicle secretions might support EPM formation because vesicles could include or attach to proteins, including chaperons, contributing to EPM formation.” in page 11 line 294-296 in the revised manuscript.

Comment#2: Lines 127-131. This description claims the appearance of a matrix of extracellular fibers which are not visible. “Intracellular” should be “intercellular”? Perhaps the description of the matrix should be deferred to Fig 3. What is the vertical “plate”?

Response: The reviewer's comment is correct.We have changed from “Intracellular” to “intercellular”in page 4 line 128 in the revised manuscript. We have put “intercellular” in the sentence “The intercellular space was occupied by a matrix containing thin extracellular fibers (3-7 nm in diameter) interspersed with secreted vesicles (Fig. 3c).” in page 4 line 144-145 in the revised manuscript.

In accordance with the reviewer’s comment, we have removed the follow sentences “, and the intracellular space was mainly occupied by a surrounding matrix containing thin extracellular fibers (3-7 nm in diameter) (Fig. 2). All these results suggested biofilm formation.”4 line 129-130 in the previous manuscript.

We have changed from “The intercellular space was occupied by a matrix containing thin extracellular fibers  interspersed with secreted vesicles (Fig. 3c).” to “The intercellular space was occupied by a matrix containing thin extracellular fibers (3-7 nm in diameter) interspersed with secreted vesicles (Fig. 3c).” in page 4 line 144-145 in the revised manuscript.

We have added the following sentence “All these results suggested biofilm formation.” in page 4 line 151 in the revised manuscript.

We have removed “vertical plate”in page 3 line 130 in the previous manuscript.

Comment#3: Lines 230-33 and 252-255. The interpretation that the mutant OM is smoother is not readily observable from the images provided.

Response: In accordance with the reviewer’s comment, we have changed from “The outer membrane of the washed cells from the DsprBcolony was smoother along the long axis of the cell (Fig. 8, right panel) than that of the WT colony (Fig. 8a, left panel), but the smoothness of the DsprBalong the short axis was similar to that of the WT (Fig. 8b)” to “The remaining outer membrane surrounding the WT cells was undulated (Fig. 8, left), as observed in Fig.7. Such undulation ofouter membrane was also observed forthe cells from DsprBcolony (Fig. 8a, right).”in page 10 line 255-257 in the revised manuscript.

We have also changed Fig. 8 legend from “TEM images of surface coat and membrane of WT and DsprBcells isolated from their colonies formed on the agar. Cells were isolated from a few mm-zone from the edges of WT and DsprBcolonies formed on 1% A-PY2, washed with the washing buffer, and aldehyde fixed. After dehydration, the cells were embedded in Epon. The samples were 70 nm thick thin-sectioned, and imaged by TEM. (a) Thin-sections along the long axis of cells. The outer membranes of DsprBcells were smoother than those of WT. (b) Thin-sections along the short axis of cells. The coats were similarly smooth for both groups.” to “TEM images of buffer-washed WT and DsprBcells isolated from their colonies formed on the agar. Cells were isolated from a few mm-zone from the edges of WT and DsprBcolonies formed on 1% A-PY2, washed with the washing buffer, and aldehyde fixed. After dehydration, the cells were embedded in Epon. The samples were 70 nm thick thin-sectioned, stained, and imaged by TEM. (a) Thin-sections along the long axis of cells. (b) Thin-sections along the short axis of cells. After the washing, surface coat structure was not observed. The undulations of the remaining outer membrane were still observed for both WT and DsprBcells.” in page 10 line 243-249 in the revised manuscript.

In the above change, we have removed the follow sentences “The outer membrane of the cells in the DsprBcolony was smoother along the long axis of the cell (Fig. 8, right panel) than that of the WT colony (Fig. 8a, left panel), but the smoothness of the DsprBcolony along the short axis was similar to that of the WT colony (Fig. 8b).” in page 8 line 230-233 in the previous manuscript.

Comment#4: A very interesting feature of the biofilms are the migration tracks devoid of EPM. Are similar structures observed in biofilms produced by other motile bacteria? Please discuss. Also, the idea in line 282-3 that EPM secretion was preceded by migration needs to be clarified. It might be just the language, but the results suggest that EPM secretion happens first and when the bacteria glide through it, the EPM gets pushed aside? More detail on this and perhaps a figure describing the model would be helpful.

Response: Thank you for your comment. In accordance with the reviewer’s comment, we have changed from “Because gliding motility is a movement that allows bacteria to stay in contact with the solid surface, it appears that F. johnsoniaecells, partly with the assistance of SprB, can glide on the agar surface while secreting EPM, thus extending the biofilm.” to “Because gliding motility is a movement that allows bacteria to stay in contact with the solid surface, it appears that F. johnsoniaecells, partly with the assistance of adhesins such as SprB, can glide on the agar surface covered with secreted EPM while secreting vesicles, thus extending the biofilm.”in page 10 line 288-291 in the revised manuscript. Thank you for your advice to construct a graphic model. However, the results are still limited to construct it.

Comment#5: Previous studies of FJ ultrastructure should be included in the introduction. The description of individual cells, such as their undulating surface must be compared to previous literature including cryo-EM studies which preserve the native ultrastructure (e.g Liu et al, J Bacteriol. 2007 Oct;189(20):7503-6.). Are the undulations real or is it possible that they are an artefact of the EM procedure that include dehydration?

Response: In accordance with the reviewer’s comment, we have added the following sentences “SprB filaments were observed on WT cells using ammonium molybdate staining and transmission electron microscopy (TEM), but were not present on sprBmutant (DsprB) cells [3].” in page 1-2 line 43-45 in the Introduction of the revised manuscript.

We have also added the following sentences “Recently, cryo-electron tomography revealed filaments and outer membrane-associated patches near the base of the outer membrane of WT cells, but these were not observed when the nonmotile gldFmutant cells were similarly examined [5].” in page 2 line 47-49 in the introduction of the revised manuscript.

We have added a new reference, as follows:

  • Reference 5: Liu J, McBride MJ. Cell surface Filaments of the gliding bacterium Flavobacterium johnsoniae revealed by cryo-electron tomography. J Bacteriol. 2007, 189:7503-7506. in page 13 line 386-387 in the introduction of the revised manuscript.

We have also added the following sentences “The undulations observed for the layered surface coats of F. johnsoniaecells in biofilm has a possibility to be artefacts due to the dehydration pretreatment of the Epon-embedded TEM. It should be also addressed using the liquid-phase ASEM.” in page 11 line 319-322 in the revised manuscript.

Comment#6: The conclusions in lines 27-29 and 79-83 should be made more consistent with the results and discussion. The differences observed between WT and mutant are not consistently reported.

Response: The reviewer's comment is correct.We have changed from “In the nonspreading DsprBcolonies, cells were tightly packed into thick layers and surrounded by less matrix, indicating immature biofilm. We conclude that F. johnsoniaecells spread and develop biofilms by using gliding motility.” to “EPM-free channels were formed in upward biofilm protrusions,probably for cell migration. In the nonspreading DsprBcolonies, cells were tightly packed in layers and the intercellular space was occupied by less matrix, indicating immature biofilm. This result suggests that SprB is not necessary for biofilm formation. We conclude that F. johnsoniaecells use gliding motility to spread and maturate biofilms.”in page 1 line 26-30 in the revised manuscript.

We have also changed from “In DsprBcolonies, which did not spread, the cells were tightly packed and surrounded by fewer intercellular substances, including vesicles, compared to WT colonies.” to “In non-spreading DsprBcolonies, the cells were tightly packed in layers and the intercellular space was occupied by less matrix, indicating immature biofilm.”in page 2 line 85-86 in the revised manuscript.

Minor points

Comment#1: The EM methodology needs to be reviewed. There are several terms like nanogold, ASEM and immunolabeling under “4.2 fixation” which don’t appear to be used in the actual study. Therefore, the actual method used is very unclear.

Response: Thank you very much for your kindness. We have changed the “4.2 fixation” as the reviewer commented.We have also changed from “Spreading colonies on agar medium and cultured bacterial cells were fixed with 1% paraformaldehyde (PFA) and 3.5% glutaraldehyde (GA) in 0.1 M phosphate buffer (PB) (pH 7.4) for 30 min at room temperature (RT) for heavy metal staining and charged-Nanogold labeling for ASEM (biofilms were fixed with 4% PFA for 10 min at RT for immunolabeling for OM). After labeling, colonies were further fixed with 1% GA for 10 min at RT, after which the contrast of the labels/stain was increased by Nanogold labeling enhancement and/or counterstaining with heavy metals as described previously for ASEM [24]. For Epon embedding and TEM, biofilms were fixed with 2.5% GA in PB at RT for 1 h and further with 1% osmic acid (OA) in the same buffer at 4°C for 1 h.” to“For Epon embedding and TEM, spreading colonies on agar medium and cultured bacterial cells were fixed with 1% paraformaldehyde and 3.5% glutaraldehyde in 0.1 M phosphate buffer (PB) (pH 7.4) at room temperature (RT) for 3 h and further with 1% osmic acid in the same buffer at 4°C for 1 h.”in page 12 line 344-347 in the revised manuscript.

Comment#2: The gldK mutant is not used in this study – please remove from acknowledgement

Response: The reviewer's comment is correct.We have removed “gldKmutant” from acknowledgement in page 13 line 362 in the previous manuscript.

Comment#3: The method for quantifying vesicles (for Fig 3d, 4c) should be described in the methods

Response: In accordance with the reviewer’s comment, we have added the following sentences “4.5. Quantification of vesicles. The number of vesicles attached to the outer circumference of bacteria was manually counted and divided by the outer circumference to get vesicle density (vesicles/mm). The circumference of the cells was measured using ImageJ (National institute of Health). Thirty-three cells were analyzed for each region.” in page 12 line 357-361 in the methods of the revised manuscript.

Comment#4: Fig 4a. It is unclear where the three main images come from. There are lines for the top and bottom image suggesting that the top image comes from the top-most image that has bacteria in section II, while the bottom-most image appears to come from the lowest image in section III. Is that correct? But where does the middle image come from? It is also confusing that there are two labels “Top surface” and “furthest from agar surface” – are these describing the same location? Also, the axis along the surface of the agar can be better labeled.

Response: The reviewer's comment is correct.

We have improved the Figure 4a. The top-most image is enlargement of the image in section II. The top and middle is the enlargements of the images in section III. In the image in section III, the area of the top and middle images of Fig. 4a was shown.

We have added the following sentences “Top panel: (a1’) Enlarged image of the area (a1) in Fig. 4a.” in page 6 line 184-185 in the revised manuscript.

We have changed the label from “furthest from agar surface” to “Top” in the Figure 4in page 6 line 179 in the revised manuscript.

Comment#5: The abbreviation for micro (mu) has been replaced by alpha in many of the figures.

Response: Thank you very much for your kind advice. In accordance with the reviewer’s comment, we have corrected the fonts in page 3 line 120 in the revised manuscript.

Comment#6: Fig 5. The inset labels should just be “a”, “b”, “c” and “d” as they refer to Fig 6 not Fig 5.

Response: The reviewer's comment is correct.We have changed from Fig. 5a”, “Fig. 5b”, “Fig. 5c”and “Fig. 5d” to “a, “b”, “c” and “d” in the revisedFigure 5in page 7 line 206 in the revised manuscript.

Comment#7: Line 305-6. Dead cells are mentioned here in the discussion but were not mentioned in the original description of the figures. How is it known that they are dead?

Response: In accordance with the reviewer’s comment, we have removed the following sentence “Death ofcells was observed in the tightly packed region of the F. johnsoniaeDsprBmutant colony that grew on 1% A-PY2 (Figs. 5, 6, the base layers). Such death might reflect social apoptosis, including activities forming cell-free channels, to allow external nutrition to diffuse into the biofilm. Consequently,” in page 11 line 305-308 in the previous manuscript.

Comment#9: Line 258. The meaning of this statement is unclear “For F. johnsoniae, this process was affected by the cell surface conditions”

Response: In accordance with the reviewer’s comment, we have changed from “For F. johnsoniae, this process was affected by the cell surface conditions (Figs. 1-5).” to “For F. johnsoniae, this process was affected by the cell surface components and environmental factors, such as the nutrient supply and moisture (Figs. 1-5) [21].”in page 10 line 260-261 in the revised manuscript.

Comment#10: The last phrase in lines 315-6 needs revising.

Response: The reviewer's comment is correct.We have changed from “The results reported here shed light into the liquid-phase biofilm structures formed at the interfaces between air and water, and also between water and solid substrate, and accelerating study of such biofilms using various Ems.” to “The results reported here shed light into the liquid-phase biofilm structures formed at the interfaces between air and water, and also between water and solid substrate. The various electron microscopy and optical microscopy techniques used in this studies will accelerate study of such biofilms.”in page 11 line 323-326 in the revised manuscript.

Comment#11: Reference #24 is currently missing details of journal, submission status etc

Response: In accordance with the reviewer’s comment, we have changed from “24. Sato K, Naya M, Hatano Y, Kondo Y, Sato M, Narita Y, Nagano K, Naito M and Sato C. Biofilm Spreading by the Adhesin-Dependent Gliding Motility of Flavobacterium johnsoniae. 2. Surface Structure of Biofilm.”

to

“25. Sato K, Naya M, Hatano Y, Kondo Y, Sato M, Narita Y, Nagano K, Naito M and Sato C. Biofilm Spreading by the Adhesin-Dependent Gliding Motility of Flavobacterium johnsoniae. 2. Surface Structure of the Biofilm. submitted.”in page 14 line 439-441 in the revised manuscript.

Comment#12: Reference #20 is currently missing details of submission status

Response: In accordance with the reviewer’s comment, we have changedfrom “20. Sato K, Naya M, Hatano Y, Kondo Y, Sato M, Narita Y, Nagano K, Naito M, Nakayama K and Sato C.Colony spreading of the gliding bacterium Flavobacterium johnsoniae in the absence of the motility adhesin SprB. Sci. Rep. 2020”

to

“21. Sato K, Naya M, Hatano Y, Kondo Y, Sato M, Narita Y, Nagano K, Naito M, Nakayama K and Sato C.Colony spreading of the gliding bacterium Flavobacterium johnsoniae in the absence of the motility adhesin SprB. Sci. Rep. 2021, 11:697”in page 14 line 429-431 in the revised manuscript. 

We have also changed from ‘Department of Microbiology & Molecular Genetics, 6154 Biomedical Physical Sciences, Michigan State University, East Lansing, MI 48824’ to ‘Department of Clinical and Diagnostic Sciences, School of Health Sciences, Oakland University, 433 Meadow Brook Road, Rochester, MI 48309’ (revised manuscript, page 1, line 12).

We have also changed from ‘F. johnsoniaehas been observed to spread on nutrient-poor 1% agar-PY2, forming a thin film-like colony. We used electron microscopy and time-lapse fluorescence microscopy to investigate the structure of colonies formed by wild-type (WT) F. johnsoniaeand by an sprBmutant (DsprB). In both cases, the bacteria were buried in the extracellular matrix covering the top of the colony. In the spreading WT colonies, a thick fiber framework and vesicles were observed, revealing the formation of a biofilm, which is probably required for the spreading movement. At the leading edge of the WT colonies, specific paths followed by bacterial clusters were observed, and abundant vesicle secretion and subsequent matrix formation were suggested.” to “F. johnsoniaespreads on nutrient-poor 1% agar-PY2, forming a thin film-like colony. We used electron microscopy and time-lapse fluorescence microscopy to investigate the structure of colonies formed by wild-type (WT) F. johnsoniaeand by the sprBmutant (DsprB). In both cases, the bacteria were buried in the extracellular polymericmatrix (EPM) covering the top of the colony. In the spreading WT colonies, the EPM included a thick fiber framework and vesicles, revealing the formation of a biofilm, which is probably required for the spreading movement. Specific paths that were followed by bacterial clusters were observed at the leading edge of colonies, and abundant vesicle secretion and subsequent matrix formation were suggested.” in page 1 line 19-26 to summarize within 200 words in the abstract in the revised manuscript.

We have also changed from “transmission electron microscopy (TEM)” to “TEM”.in page 2 line 82 and in page 4 line 131 in the revised manuscript.

We have also changed from “a matrix” to “an EPM” in page 2 line 82 in the revised manuscript.

High-magnification images of cells in Fig. 2 are shown in Fig. 7. Inlet is the enlargement of the square in the left panel.” in page 4 line 138-139 in the revised manuscript.

We have also changed from ‘Each cell in the bacterial layer on the agar surface had a thick coat structure (arrows), which was often in contact with the coat of the neighboring cells (arrowheads).” to “Each cell in the bacterial layer on the agar surface had a thick layered coat structure (arrows). The coat structure had regular undulations along the long and short axes, and was often in contact with the coat of the neighboring cells (arrowheads).” in page 9 line 238-241 in the revised manuscript.

We have also changed from “The coat structure was not observed when the bacterial cells isolated from the colony grown on 1% A-PY2 were washed with washing buffer (10 mM Tris-HCl pH 7.5) (Fig. 8) prior to aldehyde fixation.” in page 11 line 315-316 in the revised manuscript.” to “The coat structure was not observed when the bacterial cells isolated from the colony grown on 1% A-PY2 were washed with washing buffer (10 mM Tris-HCl pH 7.5) prior to aldehyde fixation (Fig. 8).” in page 10 line 253-255 in the revised manuscript.

We have also changed from “, as observed in the biofilms of S. aureusand P. acnesby atmospheric scanning electron microscopy (ASEM) [22, 25].” to “, as observed in the biofilms of S. aureusand P. acnesimmersed in aqueous liquid by atmospheric scanning electron microscopy (ASEM) [23, 26].” in page 11 line 315-316 in the revised manuscript.

We have also changed from “Overview of the sections examined. High-magnification images of cells pictured in Fig. 2 are shown in Fig. 7. Inlet is enlargement of the surrounded area with a square.” to “Overview of the sections examined.

We have also changed from “room temperature” to “RT” in page 12 line 349 in the revised manuscript.

Reviewer 2 Report

The author report an intresting and well structured paper about Biofilm spreading by the adhesin-dependent gliding motility of Flavobacterium johnsoniae. 

The paper need only minor revision. 

Line 31. There is a ";" in place of the "."

Line 115. Check the formatting of the title.

Line 322. Fix the table. 

Author Response

Dear Reviewer2,

International Journal of Molecular Sciences,

January 27, 2021

Manuscript ID: ijms-1058259

Thank you for your email on 7-January-2021 regarding our manuscript, “Biofilm spreading by the adhesin-dependent gliding motility of Flavobacterium johnsoniae. 1. Internal structure of biofilm”.

We are very grateful for yourowncomments and the reviewer’s comments, and would like to express our cordial thanks for the valuable suggestions for our paper. We have carefully revised the text, and send you the revised version (attached), and detailed our point-by-point responses to each issue raised. Changes in the text are indicated by yellow highlights.

We hope that the revised manuscript is now suitable for publication in the International Journal of Molecular Sciences.

Sincerely,

Keiko Sato, DDS, PhD

Department of Microbiology and Oral Infection,

Nagasaki University Graduate School of Biomedical Sciences

1-7-1 Sakamoto, Nagasaki, Nagasaki 852-8588, Japan

Tel: +81-95-819-7649. Fax: +81-95-819-7650.

Chikara Sato, Ph.D.

Health and Medical Research Institute, National Institute of Advanced Industrial Science and Technology (AIST)

Umezono 1-1-4, Tsukuba, Ibaraki 305-8568 Japan

Tel: +81-29-861-5562; Fax: +81-29-861-6478

Responses to reviewer2’s comments

First of all, we would like to express our cordial thanks to referee 2 for his/her insightful comments, which encouraged us and helped us to significantly improve the paper.

The author report an intresting and well structured paper about Biofilm spreading by the adhesin-dependent gliding motility of Flavobacterium johnsoniae.

Response: Thank you very much for your evaluation of our paper.

The paper need only minor revision.

Comment#1:Line 31. There is a ";" in place of the "."

Response: The reviewer's comment is correct.We have changed from ";" to "." in page 1 line 33 in the revised manuscript.

Comment#2:Line 115. Check the formatting of the title.

Response: Thank you very much for your kind advice.We have changed the formatting of the title to italics and changed it to 10 points in page 3 line 120 in the revised manuscript.

Comment#3:Line 322. Fix the table.

Response: Thank you very much for your kind advice.We have fixed the tablein page 11 line 333 in the revised manuscript.

We have also changed from ‘Biofilm spreading by the adhesin-dependent gliding motility of Flavobacterium johnsoniae. 1. Internal structure of biofilm.” to “Biofilm spreading by the adhesin-dependent gliding motility of Flavobacterium johnsoniae. 1. Internal structure of the biofilm.”in page 1 line 1-3 in the title in the revised manuscript.

We have also changed from ‘Department of Microbiology & Molecular Genetics, 6154 Biomedical Physical Sciences, Michigan State University, East Lansing, MI 48824’ to ‘Department of Clinical and Diagnostic Sciences, School of Health Sciences, Oakland University, 433 Meadow Brook Road, Rochester, MI 48309’ (revised manuscript, page 1, line 12).

We have also changed from ‘F. johnsoniaehas been observed to spread on nutrient-poor 1% agar-PY2, forming a thin film-like colony. We used electron microscopy and time-lapse fluorescence microscopy to investigate the structure of colonies formed by wild-type (WT) F. johnsoniaeand by an sprBmutant (DsprB). In both cases, the bacteria were buried in the extracellular matrix covering the top of the colony. In the spreading WT colonies, a thick fiber framework and vesicles were observed, revealing the formation of a biofilm, which is probably required for the spreading movement. At the leading edge of the WT colonies, specific paths followed by bacterial clusters were observed, and abundant vesicle secretion and subsequent matrix formation were suggested.” to “F. johnsoniaespreads on nutrient-poor 1% agar-PY2, forming a thin film-like colony. We used electron microscopy and time-lapse fluorescence microscopy to investigate the structure of colonies formed by wild-type (WT) F. johnsoniaeand by the sprBmutant (DsprB). In both cases, the bacteria were buried in the extracellular polymericmatrix (EPM) covering the top of the colony. In the spreading WT colonies, the EPM included a thick fiber framework and vesicles, revealing the formation of a biofilm, which is probably required for the spreading movement. Specific paths that were followed by bacterial clusters were observed at the leading edge of colonies, and abundant vesicle secretion and subsequent matrix formation were suggested.” in page 1 line 19-26 to summarize within 200 words in the abstract in the revised manuscript.

We have also changed from “transmission electron microscopy (TEM)” to “TEM”.in page 2 line 82 and in page 4 line 131 in the revised manuscript.

We have also changed from “a matrix” to “an EPM” in page 2 line 82 in the revised manuscript.

High-magnification images of cells in Fig. 2 are shown in Fig. 7. Inlet is the enlargement of the square in the left panel.” in page 4 line 138-139 in the revised manuscript.

We have also changed from ‘Each cell in the bacterial layer on the agar surface had a thick coat structure (arrows), which was often in contact with the coat of the neighboring cells (arrowheads).” to “Each cell in the bacterial layer on the agar surface had a thick layered coat structure (arrows). The coat structure had regular undulations along the long and short axes, and was often in contact with the coat of the neighboring cells (arrowheads).” in page 9 line 238-241 in the revised manuscript.

We have also changed from “The coat structure was not observed when the bacterial cells isolated from the colony grown on 1% A-PY2 were washed with washing buffer (10 mM Tris-HCl pH 7.5) (Fig. 8) prior to aldehyde fixation.” in page 11 line 315-316 in the revised manuscript.” to “The coat structure was not observed when the bacterial cells isolated from the colony grown on 1% A-PY2 were washed with washing buffer (10 mM Tris-HCl pH 7.5) prior to aldehyde fixation (Fig. 8).” in page 10 line 253-255 in the revised manuscript.

We have also changed from “, as observed in the biofilms of S. aureusand P. acnesby atmospheric scanning electron microscopy (ASEM) [22, 25].” to “, as observed in the biofilms of S. aureusand P. acnesimmersed in aqueous liquid by atmospheric scanning electron microscopy (ASEM) [23, 26].” in page 11 line 315-316 in the revised manuscript.

We have also changed from “Overview of the sections examined. High-magnification images of cells pictured in Fig. 2 are shown in Fig. 7. Inlet is enlargement of the surrounded area with a square.” to “Overview of the sections examined.

We have also changed from “room temperature” to “RT” in page 12 line 349 in the revised manuscript.

Round 2

Reviewer 1 Report

Line 47: The new reference is not recent. Delete "recently"

Line 139: "inlet" should be "inset"

Line 285: I think this is a language problem. I think you mean to say that EPM secretion happens first, and then the movement of cells. Therefore, please say "These observations suggest that secretion of EPM preceded the migration of F. johnsoniae cells"

All other points have been adequately addressed by the authors

Author Response

Responses to reviewer1’s comments

First of all, we would like to express our cordial thanks to referee 1 for his/her insightful comments, which encouraged us and helped us to significantly improve the paper.

Comment#1: Line 47: The new reference is not recent. Delete "recently"

Response: In accordance with the reviewer’s comment, we have removed “recently" in page 2 line 47 in the revised manuscript.

Comment#2: Line 139: "inlet" should be "inset"

Response: Thank you so much for your comment. In accordance with the reviewer’s comment, we have changed from “inlet" to "inset" in page 4 line 137 in the revised manuscript.

Comment#3: Line 285: I think this is a language problem. I think you mean to say that EPM secretion happens first, and then the movement of cells. Therefore, please say "These observations suggest that secretion of EPM preceded the migration of F. johnsoniae cells"

Response: Thank you very much for your comment. In accordance with the reviewer’s comment, we have changed from “These observations suggest that secretion of EPM was preceded by the migration of F. johnsoniae cells.” to “These observations suggest that secretion of EPM preceded the migration of F. johnsoniae cells.” in page 10 line 281-282 in the revised manuscript.

Response: In accordance with the editor’s comment, we have removed the follow references,

“8. Sato K, Sakai E, Veith PD, Shoji M, Kikuchi Y, Yukitake H, Ohara N, Naito M, Okamoto K, Reynolds EC and Nakayama K. Identification of a new membrane-associated protein that influences transport/maturation of gingipains and adhesins of Porphyromonas gingivalis. J. Biol. Chem. 2005, 280: 8668-77.

  1. Sato K, Yukitake H, Narita Y, Shoji M, Naito M, Nakayama K. Identification of Porphyromonas gingivalis proteins secreted by the Por secretion system. FEMS Microbiol Lett. 2013, 338(1):68-76.
  2. Paul D Veith, Nor A Nor Muhammad, Stuart G Dashper, Vladimir A Likić, Dhana G Gorasia, Dina Chen, Samantha J Byrne, Deanne V Catmull, Eric C Reynolds. Protein substrates of a novel secretion system are numerous in the Bacteroidetes phylum and have in common a cleavable C-terminal secretion signal, extensive post-translational modification, and cell-surface attachment. J Proteome Res. 2013, 12:4449-61.
  3. Maruyama Y, Ebihara T, Nishiyama H, Konyuba Y, Senda M, Numaga-Tomita T, Senda T, Suga M, Sato C, Direct observation of protein microcrystals in crystallization buffer by atmospheric scanning electron microscopy. Int. J. Mol. Sci. 2012, 13, 10553-10567.
  4. Nishiyama H, Teramoto K, Suga M, Sato C. Positively charged nanogold label allows the observation of fine cell filopodia and flagella in solution by atmospheric scanning electron microscopy. Microsc Res Tech. 2014, 77:153-60.
  5. Murai T, Sato M, Nishiyama H, Suga M, Sato C. Ultrastructural analysis of nanogold-labeled cell surface microvilli in liquid by atmospheric scanning electron microscopy and their relevance in cell adhesion. Int. J. Mol. Sci. 2013, 14, 20809-20819.
  6. Yamazawa T, Nakamura N, Sato M, Sato C. Secretory glands and microvascular systems imaged in aqueous solution by atmospheric scanning electron microscopy (ASEM) Microsc Res Tech. 2016, 79, 1179-1187.
  7. Komenami T, Yoshimura A, Matsuno Y, Sato M, Sato C. Network of palladium-based nanorings synthesized by liquid-phase reduction using DMSO-H2O: In situ monitoring of structure formation and drying deformation by ASEM. Int. J. Mol. Sci. 2020, 21, 3271.” in the previous manuscript.